# A Systematic Review and Meta-Analysis of Anti-Rheumatic Drugs and Pneumococcal Vaccine Immunogenicity in Inflammatory Arthritis

**DOI:** 10.3390/vaccines11111680

**Published:** 2023-11-02

**Authors:** Deepak Nagra, Katie Bechman, Maryam Adas, Zijing Yang, Edward Alveyn, Sujith Subesinghe, Andrew Rutherford, Victoria Allen, Samir Patel, Mark D. Russell, Andrew Cope, Sam Norton, James Galloway

**Affiliations:** Centre for Rheumatic Disease, King’s College London, London WC2R 2LS, UKedward.alveyn@nhs.net (E.A.); samir.patel15@nhs.net (S.P.);

**Keywords:** rheumatoid arthritis, pneumococcal vaccination, immunosuppression

## Abstract

Background: Pneumococcal pneumonia is an important cause of morbidity and mortality amongst patients with inflammatory arthritis. Vaccination is recommended by the National Institute for Health and Care Excellence (NICE) but it remains unclear how vaccine efficacy is impacted by different immunosuppressive agents. Our objective was to compare the chance of a seroconversion following vaccination against pneumococcus in patients with inflammatory arthritis to that in the general population, as well as to compare the chance of seroconversion across different targeted therapies. Methods: We searched MEDLINE, Embase and the Cochrane Library databases from inception until 20 June 2023. We included randomized controlled trials and observational studies. Aggregate data were used to undertake a pairwise meta-analysis. Our primary outcome of interest was vaccine seroconversion. We accepted the definition of serological response reported by the authors of each study. Results: Twenty studies were identified in the systematic review (2807 patients) with ten reporting sufficient data to be included in the meta-analysis (1443 patients). The chance of seroconversion in patients receiving targeted therapies, relative to the general population, was 0.61 (95% CI 0.35 to 1.08). The reduced odds of response were skewed strongly by the effects of abatacept and rituximab with no difference between patients on TNF inhibitors (TNFis) or IL-6 inhibition and healthy controls. Within different inflammatory arthritis populations the findings remained consistent, with rituximab having the strongest negative impact on vaccine response. TNF inhibition monotherapy was associated with a greater chance of vaccine response compared with methotrexate (2.25 (95% CI 1.28 to 3.96)). JAK inhibitor (JAKi) studies were few in number and did not present comparable vaccine response endpoints to include in the meta-analysis. The information available does not suggest any significant detrimental effects of JAKi on vaccine response. Conclusion: This updated meta-analysis confirms that, for most patients with inflammatory arthritis, pneumococcal vaccine can be administered with confidence and that it will achieve comparable seroconversion rates to the healthy population. Patients on rituximab were the group least likely to achieve a response and further research is needed to explore the value of multiple-course pneumococcal vaccination schedules in this population.

## 1. Introduction

Inflammatory arthritides such as rheumatoid arthritis (RA) are common autoimmune diseases affecting approximately 1% of the adult population [1]. Ankylosing spondylitis, psoriatic arthritis, enteropathic arthritis and undifferentiated inflammatory arthritis treatments have advanced in recent times with the use of immunomodulating agents. Contemporary therapies have radically altered the outcomes for RA through targeted immune modulation (such as tumour necrosis factor inhibitors [TNFis]) and prevention of joint damage. However, an ongoing concern for patients on immune modulation is risk of infection [2]. It is unclear whether newer therapies such as Janus kinase inhibitors (JAKi), IL-6 inhibitors (tocilizumab), IL-17 inhibitors (e.g., ixekizumab and secukinumab) and IL-23 inhibitors (e.g., ustekinumab) have similar or differing infection risk profiles.

Pneumococcal pneumonia and invasive pneumococcal disease (IPD) represent a rare but very serious infection that is recognized as more common amongst RA patients [3]. In addition, RA patients are at risk of pneumococcal septic arthritis [4]. Vaccination against pneumococcal disease represents a mitigation strategy which is recommended by guidelines around the globe including in Europe and the US [3,5]. Pneumococcal vaccine uptake, however, is low amongst RA patients [6].

The aim of this study is to conduct a revised systematic literature review and meta-analysis of randomized controlled trials and observational studies describing pneumococcal vaccine response amongst people with inflammatory arthritis receiving immunosuppression that describe pneumococcal vaccine response.

## 2. Methods

### 2.1. Data Sources and Eligibility Criteria

MEDLINE Embase and Cochrane Library databases were searched from inception until 28 June 2023 for studies describing pneumococcal vaccine response in patients with inflammatory arthritis receiving targeted immune modulation with a comparator group of either standard disease-modifying anti-rheumatic drug (DMARD) therapies or healthy controls. The selection criteria were limited to studies that were either randomized controlled trials (RCT) or cohort studies (prospective and retrospective). Other study designs were excluded such as case reports or case series. Studies were excluded if there were inadequate outcome data, a drug or disease that was not being studied as part of this review (e.g., belimumab and systemic lupus erythematous). Only studies published in full were eligible. The full search strategy is available in the Appendix A. The study design was published a priori on the International Prospective Register of Systematic Reviews (CRD42023363018). The analysis was performed in accordance with the Preferred Reporting Items for Systematic Reviews and Meta-Analysis (PRISMA) [7].

### 2.2. Data Extraction

Study titles and abstracts were screened independently by four investigators (DN, EA, SP, and MA) and full tests of relevant studies were assessed for eligibility. Disagreements were resolved through the involvement of additional reviewers (JG) if required.

### 2.3. Outcomes

The primary outcome was pneumococcal vaccine immunogenicity as defined by the authors of each individual study and included both cellular and humoral response. Secondary analyses were conducted, limiting the studies to those that (i) used a definition of vaccine immunogenicity as a two-fold change in vaccine serotype IgG; (ii) an absolute increase in vaccine serotype titre of 0.35 mcg/mL.

### 2.4. Bias and Quality Assessment

The Newcastle Ottawa Scale (NOS) was used for non-randomized studies. The NOS and Cochrane risk-of-bias tool were used for RCTs [8]. The assessments were conducted by two reviewers (DN and MA) and discrepancies were resolved through discussion with a third reviewer (JG).

### 2.5. Data Synthesis and Analysis

Data analysis was undertaken using STATA 17 and R version 4.2.2. Baseline characteristics for eligible studies were described in tabular format without inferential statistics. Odds ratios (OR) with 95% confidence intervals were calculated for each study arm comparison. Pairwise meta-analysis was undertaken to estimate pooled ORs with 95% confidence intervals of vaccine response between comparators. Pairwise meta-analysis was conducted using the random-effects maximum likelihood method and presented graphically with forest plots. Between-study heterogeneity was assessed using the I^2^ and Cochrane Q statistics. If a study had no patients responding to the vaccine in one or more arm then a treatment arm continuity correction was applied. The pairwise analyses were run separately for comparisons between (i) general population and patients on targeted immunosuppression; (ii) patients on targeted immunosuppression in combination with methotrexate compared to patients on methotrexate alone; (iii) patients on targeted immunosuppression monotherapy compared to methotrexate monotherapy.

Where studies did not provide absolute definitions of vaccine immunogenicity, a narrative summary was used.

A two-sided *p* value of less than 0.05 was considered significant. No adjustment for multiplicity was undertaken to minimize the risk of a type 2 error.

## 3. Results

### 3.1. Study Characteristics

The literature search yielded 4234 articles. Following screening for eligibility, 20 studies were included (Figure 1). Ten studies had a healthy control comparator and ten had a DMARD control. The total number of patients studied through the 20 reviews was 2807 with 1443 included in the meta-analysis. Of the 20 studies, 3 included rituximab, 8 included TNFis, 5 included IL-6 inhibitors, 3 included abatacept, 2 included JAKi and 14 included methotrexate. Some studies included non-RA patients and non-RA licensed therapies (ocrelizumab). The data from those patients were excluded from the analysis. Seventeen studies had patients with rheumatoid arthritis alone, one study included patients with inflammatory bowel disease, one study included patients with psoriatic arthritis and one study included patients with rheumatoid, psoriatic arthritis and axial spondyloarthropathies. The pneumococcal polysaccharide 23 valent (PPVS23) vaccine was used in 16 studies, the pneumococcal conjugate 13 (PV13) in two studies, the PCV-7 in 1 study and both PPVS 23 and PCV13 combined was used in two studies.

### 3.2. Risk of Bias and Publication Bias

Of the 20 included studies, 19 were at low risk of bias and 1 was at moderate risk. Of the studies, seven were randomized trials with all seven were at low risk of bias. The full risk of bias scoring is presented in the Appendix B (Table A1). There was no evidence of publication bias for the primary outcome. Based on no studies being identified at high risk of bias, there was insufficient justification for sensitivity analyses excluding any studies following the risk of bias assessment.

### 3.3. Primary Outcome

(i) general population and patients with inflammatory arthritis on targeted immunosuppression;

The primary outcome of pneumococcal vaccine response was observed in 71% (2001/2807) of patients with rheumatic disease on immunosuppression compared with 61% (241/393) of healthy controls. The pooled OR for a vaccine response was 0.61 (95% CI [0.35 to 1.08]). The heterogeneity statistic (I^2^) was 56% indicating moderate heterogeneity (20 studies). Furthermore, the Cochrane Q test of heterogeneity was significant (*p* = 0.02). This was expected given the different treatment classes being considered. Therefore, differences in effect were considered by drug class; the largest negative effect on vaccine response relative to healthy controls was observed for rituximab (OR 0.14 (95% CI [0.03 to 0.54])) followed by abatacept (OR 0.50 (95% CI [0.24 to 1.07]). The estimates for tocilizumab and TNFis suggested no difference in the odds of vaccine response when compared to healthy controls.

(ii) patients with inflammatory arthritis on targeted immunosuppression in combination with methotrexate, compared to patients with inflammatory arthritis on methotrexate alone.

The pairwise comparison between biologic and methotrexate combination therapy compared to methotrexate monotherapy showed no overall significant association with vaccine response and there was less evidence of heterogeneity (Cochrane Q test *p* = 0.42). The odds ratio for response was numerically lowest for the rituximab cohort, although this difference was not statistically different (odds ratio 0.61 (95% CI 0.23 to 1.61)). The forest plot of estimate is shown in Figure 2.

Focusing on biologic monotherapy compared to methotrexate, the differences were of greater magnitude driven by an increased odds ratio for vaccine response amongst TNF users (odds ratio 2.25 (95% CI 1.28 to 3.96)). The estimate for tocilizumab was also in favour of tocilizumab compared to methotrexate (odds ratio 1.31 [0.51 to 3.37]), whilst the effect for abatacept was neutral compared to methotrexate. Rituximab appeared deleterious to vaccine response albeit based on a single study. The forest plot of estimate is shown in Figure 3.

From the 20 studies, 13 included a definition of a two-fold change in titre, of which 10 were within the meta-analysis. None of the studies included a definition based on an absolute increase in vaccine serotype titre of 0.35 mcg/mL. The sensitivity analysis limiting studies to those that reported a two-fold change demonstrated consistent findings and there were too few studies to compare responses based upon absolute changes in vaccine serology.

### 3.4. Detailed Description of Individual Drugs Incorporating Narrative Synthesis

#### 3.4.1. TNF Inhibitors

Five studies included patients on TNFis (Kapetanovic et al., 2006 [9], Kaine et al., 2007 [10], Visvanathan et al., 2007 [11], Kapetanovic et al., 2011 [12] and Kivitz et al., 2014 [13]). These included two studies with infliximab [9,11] and one study with etanercept [9], adalimumab [10] and certolizumab [13]. This included eight comparisons of TNFi versus a methotrexate arm (of which five were TNFi combination therapy with methotrexate and three were TNFi monotherapy) and two comparisons of TNFi versus a healthy control arm.

TNFis did not impact the pneumococcal vaccine response when compared to healthy control (OR 1.10 [95% CI 0.70, 1.73]) (Figure 1) or compared to methotrexate (OR 1.30 [95% CI 0.88, 1.90]) (Figure 2). In comparisons that were restricted to TNFi monotherapy, TNFis was associated with a greater pneumococcal vaccine response compared to methotrexate alone (OR 2.25 [95% CI 1.28, 3.96]) (Figure 3).

#### 3.4.2. Tocilizumab

Two studies examined tocilizumab with a methotrexate comparison arm (one with tocilizumab monotherapy and one as combination therapy with methotrexate) (Kapetanovic et al., 2011 [12] and Bingham et al., 2015 [14]). There was no difference in immunogenicity when compared against methotrexate (OR 1.31, [95% CI 0.51, 3.37]). There was only one comparison against healthy control, which demonstrated similar results (OR 0.96 [95% CI 0.38, 2.40]).

#### 3.4.3. Abatacept

Abatacept was examined in two studies (Migita et al., 2015 [15] and Kapetanovic 2013 [16]). Both included a methotrexate and healthy control comparator arm and assessed abatacept monotherapy. There was no difference in vaccine response between abatacept and methotrexate (OR 0.82 [95% CI 0.39, 1.74]) and abatacept and healthy controls (OR 0.50 [95% CI 0.24, 1.07], (Figure 4)), although the estimate was in the direction of a reduced vaccine response.

#### 3.4.4. Rituximab

Two studies (Bingham et al., 2015 [14] and Kapetanovic et al., 2013 [16]) examined rituximab. These included three comparisons of rituximab versus a methotrexate arm (of which two were rituximab combination therapy and one was monotherapy) and two comparisons of rituximab versus a healthy control arm (one with rituximab combination therapy and one as monotherapy). There was no difference in immunogenicity when compared against methotrexate (OR 0.61 [95% CI 0.23, 1.61]) although the estimate was in the direction of a reduced vaccine response. Rituximab was associated with a statistically significant reduction in vaccine response when compared to healthy controls (OR 0.14 [95% CI 0.03, 0.54]).

#### 3.4.5. JAK Inhibitors

Only two studies included JAKi. Garrido et al., 2022 [17] explored immunosuppression in general and included just three patients on JAKi so was unable to explore JAKi-specific effects. Mori et al., 2023 [18] described fifty-three patients receiving immunomodulation of whom forty-three were on JAKis (twenty monotherapy, twenty-three on combination therapy) compared to ten methotrexate monotherapy patients. All the patients significantly increased IgG response irrespective of treatment but the fold increase was numerically reduced amongst the JAKi-treated cohort. Based upon their primary definition of response it was the methotrexate and JAKi combination that had the lowest response to vaccine of 52% in the methotrexate JAKi combination group, 90% in methotrexate monotherapy and 95% in JAKi monotherapy. The study included three JAKis, tofacitinib, baricitinib and peficitinib and no differences were observed between these drugs. Of note, there was a significant imbalance in background baseline steroid use as no patients in the methotrexate monotherapy group were on background prednisolone, compared to 4/20 in the JAKi monotherapy group and 7/23 in the combination group.

## 4. Discussion

From the anti-rheumatic therapies for which we had available data, rituximab and methotrexate appeared to have the greatest hindrance to pneumococcal vaccine response. This was supported by our meta-analysis as well as the narrative synthesis from studies not eligible for inclusion in the meta-analysis. There was no evidence that TNFis meaningfully alter pneumococcal vaccine response and data were insufficient on other targeted therapies to draw robust conclusions.

The strengths of this systematic review and meta-analysis include the comprehensive literature search with clearly defined eligibility criteria. Cohorts included in the meta-analysis were at low risk of bias and were comparable in terms of study design exposure and outcome. The review has also been able to capture the interplay between targeted therapy with and without methotrexate.

The findings are largely consistent with a previous meta-analysis undertaken (Subesinghe et al., 2018 [19]), although the advantages of this updated review include more precise estimates of effect based on larger numbers of studies included as well as the inclusion of early data exploring novel therapies such as JAKis.

There are some important limitations of this study. We have focused on outcomes which were comparable across studies but the study populations themselves were heterogenous with differences in the number of prior immunotherapies, disease duration and geography. There were limited numbers of studies that included JAKis. There were insufficient data to explore the different pneumococcal vaccine subtypes (polysaccharide and conjugate).

Although it may seem intuitive that any immunosuppression would blunt vaccine response this would not be correct. Immune response to vaccines is complex and depends on many parts of the immune system, but not all parts. In addition, different vaccines require different aspects of the immune system to function. A successful vaccine response will depend upon recognition of the vaccine antigen by innate immune cells such as dendritic cells. The dendritic cells produce a cascade of signalling molecules including cytokines to attract adaptative immune cells; in particular, CD4 and CD8 T cells. The T cell response in turn drives B cell maturation and antibody response. The use of non-selective immunosuppressive therapies such as corticosteroids or methotrexate would predictably impact on multiple steps in the vaccine response as these drugs have been shown to suppress both innate and adaptive immune responses.

Targeted therapies such as TNFis do not necessarily block any of the pathways required for vaccine response and it is perhaps not surprising that we have observed the results described. In contrast, B cell depletion has much more marked impacts on vaccine response, which fits with both the inhibition of the B cell as an antigen presenting cell and also the subsequent downstream development of plasma cells to permit the formation of new antibody production.

The observation that rituximab has a more detrimental impact on vaccine response when compared to other targeted therapies has been observed with influenza and COVID-19 vaccines. Randomized controlled trials have demonstrated that interrupting methotrexate around the time of vaccination against both influenza and COVID-19 improves vaccine immunogenicity (Abhishek et al., 2022 [20]). Comparable studies to test the benefit of DMARD interruption have not been performed for pneumococcal vaccines. There are important differences between pneumococcal vaccine and influenza and COVID-19 vaccines: pneumococcal vaccines are available either as a simple polysaccharide or as a conjugated protein. The conjugated vaccine enables a more potent T-cell recognition of the antigens and a stronger immunological response. Strategies that involve interrupting DMARD therapy have a risk of disease flare and a conjugate vaccine approach may be an alternative strategy to maximize vaccine response in an immunosuppressed population.

We have an ageing population. The average age of autoimmune disease onset has risen in recent years and the number of people living with long-term conditions in the community has never been greater. Risk mitigation with vaccines is an essential component of the rheumatologist’s toolkit but more work needs to be undertaken to know how to best use this. Based on the results of this systematic review it seems sensible to advocate using the most potent vaccination strategy, e.g., conjugate vaccines in patients who are currently receiving rituximab or methotrexate.

## 5. Conclusions

A key challenge for vaccinating RA patients is that current treatment paradigms advocate for early aggressive therapy and patients and clinicians are not keen to delay treatment initiation whilst awaiting vaccines. However, once treatment has started there is concern that vaccines will be less effective.

A previous systematic review was published by our group in 2018; however; multiple studies have been published since that original review, justifying an update [19].

Our meta-analysis has demonstrated that methotrexate and rituximab reduce the immunogenicity of the pneumococcal vaccine response. The addition of methotrexate to biologic agents also hinders the vaccine response. Abatacept, tocilizumab, TNFis and JAKis when used as monotherapy have not demonstrated a reduced antibody response.

## Figures and Tables

**Figure 1 vaccines-11-01680-f001:**
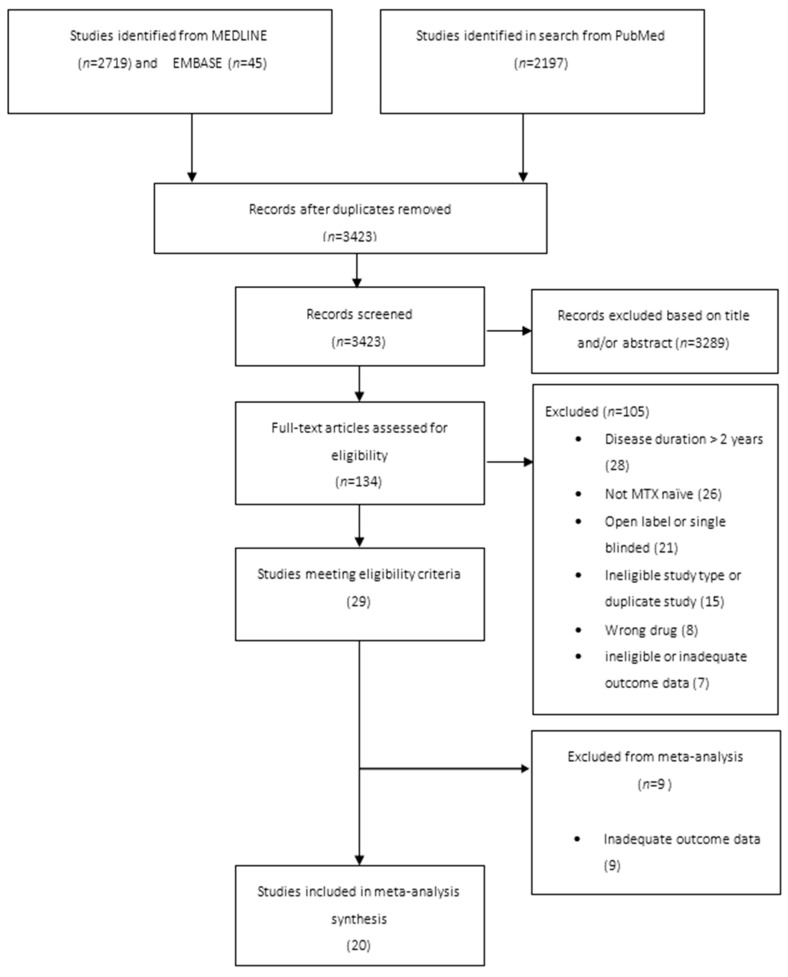
PRISMA flowchart of studies identified through the systematic literature review.

**Figure 2 vaccines-11-01680-f002:**
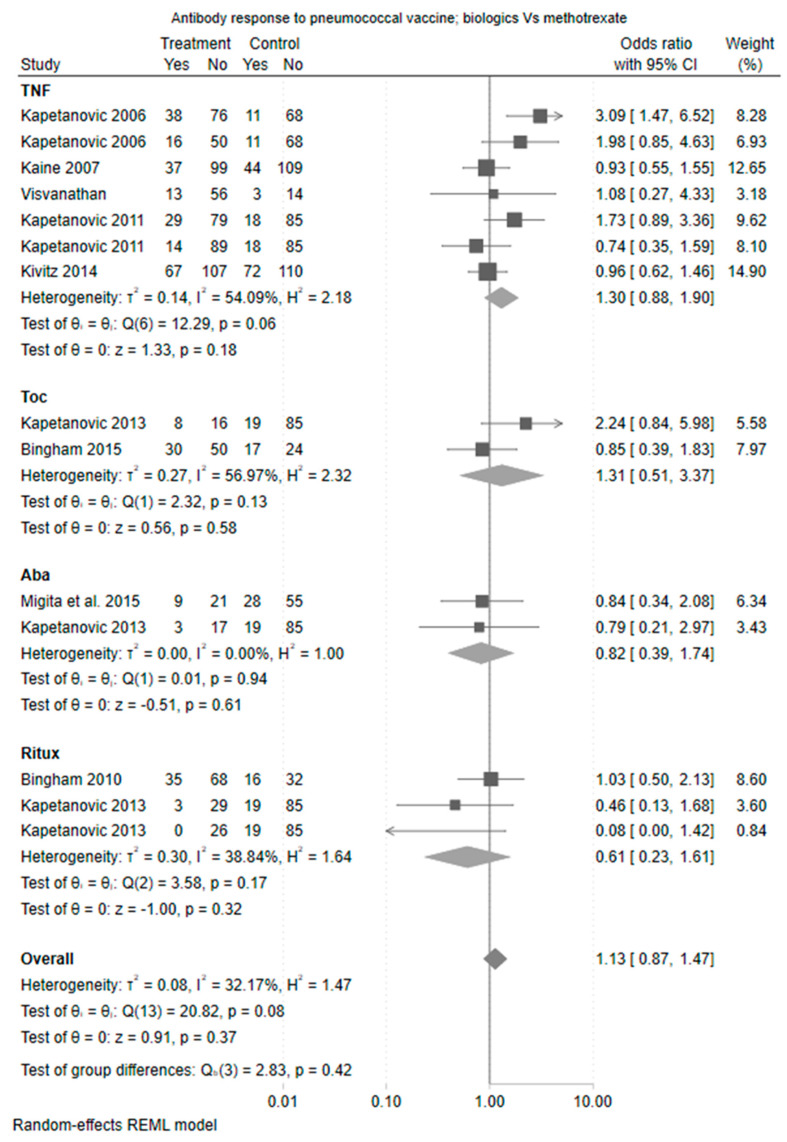
Forest plot—risk ratio of antibody response to pneumococcal vaccine between biologic monotherapy and methotrexate. Aba: Abatacept; Toc: Toculizumab; TNF: Tumor necrosis factor inhibitors; Ritux: Rituximab [9,10,11,12,13,14,15,16].

**Figure 3 vaccines-11-01680-f003:**
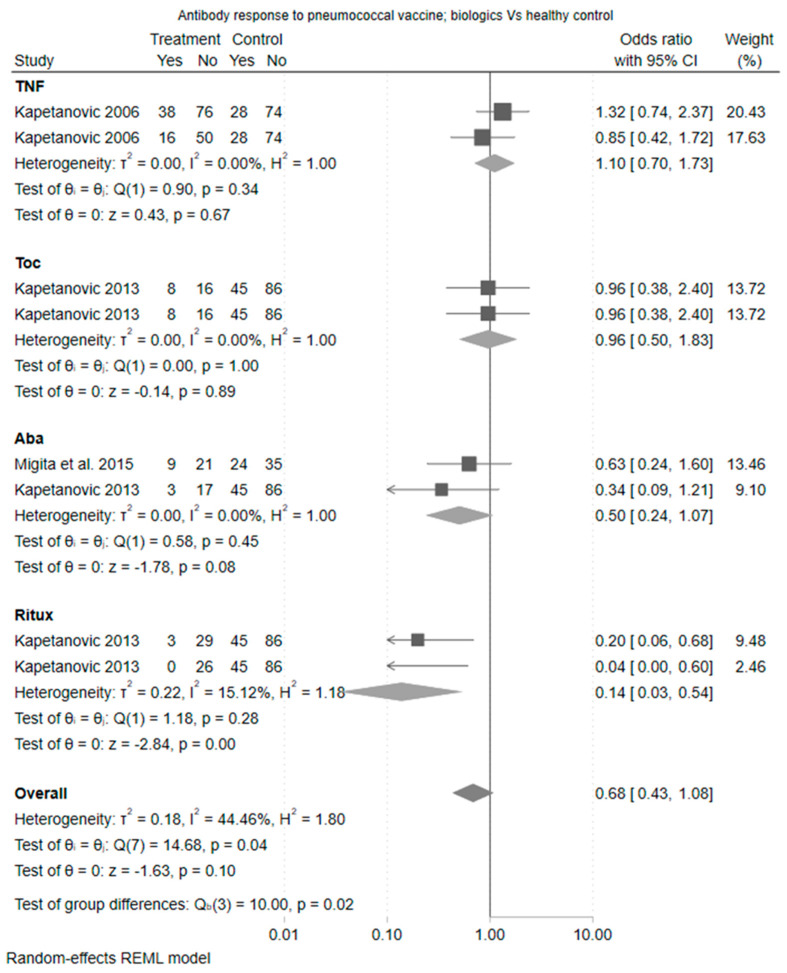
Forest plot—antibody response to pneumococcal vaccine between biologic and healthy control. Aba: Abatacept; Toc: Toculizumab; TNF: Tumor necrosis factor inhibitors; Ritux: Rituximab [9,10,11,12,13,14,15,16].

**Figure 4 vaccines-11-01680-f004:**
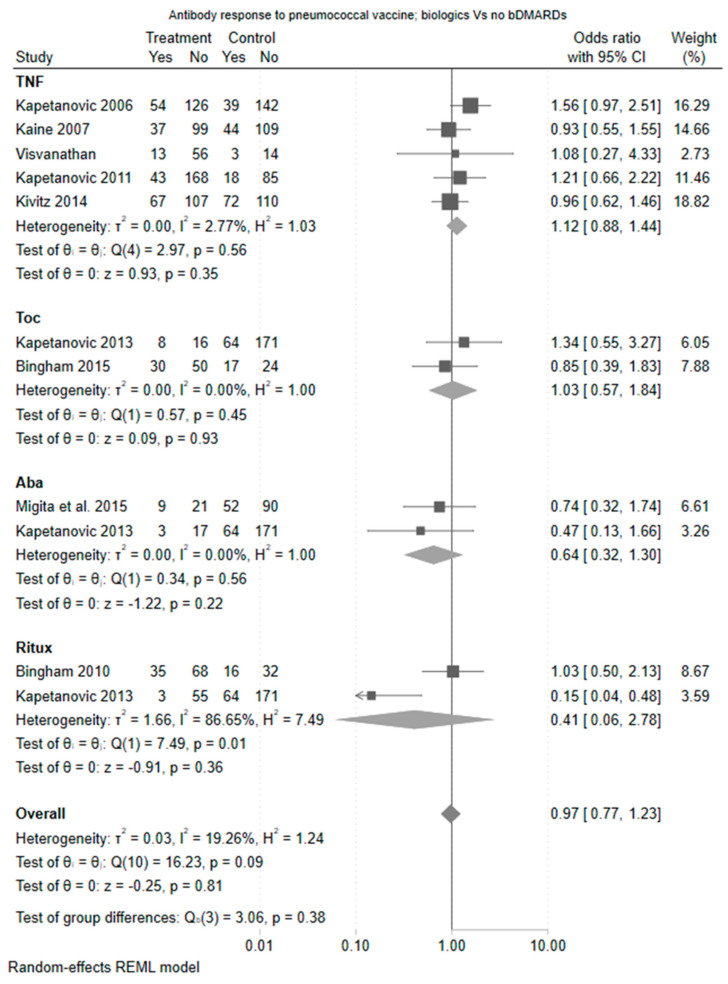
Forest plot—antibody response to pneumococcal vaccine for biologic with no csDMARDs. Aba: Abatacept; Toc: Toculizumab; TNF: Tumor necrosis factor inhibitors; Ritux: Rituximab [9,10,11,12,13,14,15,16].

## Data Availability

The data underlying this article are available from research articles in the public domain.

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
