# Peer review of "A Systematic Review and Meta-Analysis of Anti-Rheumatic Drugs and Pneumococcal Vaccine Immunogenicity in Inflammatory Arthritis"

_vaccines, 2023, doi:10.3390/vaccines11111680_

Round 1
Reviewer 1 Report
Comments and Suggestions for Authors
1. What is the main question addressed by the research?
This review manuscript systematically surveys the literature analyzing the efficacy of pneumococcal vaccines under immunosuppressive and -modulatory therapy given to patients with RA.
2. Do you consider the topic original or relevant in the field? Does it
address a specific gap in the field?
This is a thorough and convincing review update from the same group on their previous publication on antirheumatic drug and vaccine immunogenicity in RA which is a critical consideration in clinical practice.
3. What does it add to the subject area compared with other published material?
This review includes significant updates that warrant this publication, including the analysis of novel specific treatments and an extended study base.
4. What specific improvements should the authors consider regarding the methodology? What further controls should be considered?
The authors clearly and convincingly present their methods, database searches and confounders.
5. Are the conclusions consistent with the evidence and arguments presented and do they address the main question posed?
Despite its limitations regarding patient populations, treatment history, disease duration and lack to differentiate between different types of vaccines, the strengths outweigh the weaknesses of this updated review. The conclusions are in line with observations on the impact of broadly immunosuppressive drugs versus targeted immunotherapy on other vaccine efficacy and are generally expected. Nevertheless, it is important for clinical practice to establish that single treatment with abatacept, tocilizumab, TNF inhibitors and JAK inhibitors is not correlated with significantly diminished potency of pneumococcal vaccines.
6. Are the references appropriate?
The review lists relevant references and discusses well the selection approaches.
7. Please include any additional comments on the tables and figures.
The manuscript is nicely illustrated, and tables are informative supporting the conclusions.
There are some oddly phrased sentences, including,
"The dendritic cells the produce a cascade of signalling molecules including cytokines to attract adaptative immune cells in particular CD4 and CD8 T cells."
"We have an ageing population the average age of autoimmune disease onset has been rising in recent years and the number of people living with long term conditions in the 269 community has never been greater."
Author Response
Many thanks for your comments.
We have taken into consideration points 1 to 7 and thank you for your input. We have changed the wording of the text with regards to your final comment on the quality of english
Reviewer 2 Report
Comments and Suggestions for Authors
This paper is well written as a review. Here are some comments.
1. James Galloway's affiliation is unknown, so please check.
2. All abbreviations and acronyms should be spelled out the first time they are used in the body of the manuscript, followed by the abbreviation or acronym in parentheses. All subsequent uses, including tables and figures, should use the abbreviation or acronym. Of particular note is the mixing of TNFi and TNF inhibitors.
3. The introduction is short, so please expand it.
4. Please add a legend to Figure 1.
5. The exclusion criteria in Figure 1 are unclear. It would be better to include who made the decision to exclude them and how, and if there is a list of excluded papers as a supplementary table (especially for n=134 or later).
6. In 3.4.1. TNF inhibitors, please specify the type of drug (etanercept, adalimumab, infliximab, etc.). Although they are grouped together as TNF inhibitors, the mechanism of action is actually different for each TNF inhibitor.
7. In 3.4.2. IL-6 inhibitors, although IL-6 inhibitors are stated as the plural, only tocilizumab is actually mentioned. Please consider either changing the subtitle to "3.4.2. Tocilizumab" or adding information about IL-6 inhibitors other than tocilizumab.
Author Response
Many thanks for taking the time to review this manuscript.
We have taken all of your comments into consideration and changed the manuscript accordingly.
With regards to point 5 - the primsa flow chart was the wrong chart which was copied and pasted into the manuscript at the time of submission. I have gone back and checked every paper and also inserted the correct flowchart. We do apologize for the initial error.
I have increased the word count for the introduction too. I have not made it too long with only a marginal increase as i didn't want to make the content of the manuscript repetitive. If you would like me to increase the word count please do let me know.
I hope you enjoy reading the revised manuscript.

Round 2
Reviewer 2 Report
Comments and Suggestions for Authors
This paper has good revisions.